# RIO-kinase 2 is essential for hematopoiesis

**Jan-Erik Messling**[1], **Isabel Peña-Rømer**[2], **Ann Sophie Moroni**[1☯], **Sarah Bruestl**[2☯], **Kristian Helin**[2]*

1 Biotech Research and Innovation Centre, University of Copenhagen, Copenhagen, Denmark, 2 The Institute of Cancer Research, London, United Kingdom

☯ These authors contributed equally to this work.
* kristian.helin@icr.ac.uk

**Data Availability Statement:** All relevant data are within the manuscript and its Supporting Information files.

**Funding:** J.-E.M. was supported by the Novo Nordisk Foundation (NNF) Copenhagen Bioscience PhD Program (NNF18CC0033666).

## Abstract

Regulation of protein synthesis is a key factor in hematopoietic stem cell maintenance and differentiation. Rio-kinase 2 (RIOK2) is a ribosome biogenesis factor that has recently been described an important regulator of human blood cell development. Additionally, we have previously identified RIOK2 as a regulator of protein synthesis and a potential target for the treatment of acute myeloid leukemia (AML). However, its functional relevance in several organ systems, including normal hematopoiesis, is not well understood. Here, we investigate the consequences of RIOK2 loss on normal hematopoiesis using two different conditional knockout mouse models. Using competitive and non-competitive bone marrow transplantations, we demonstrate that RIOK2 is essential for the differentiation of hematopoietic stem and progenitor cells (HSPCs) as well as for the maintenance of fully differentiated blood cells *in vivo* as well as *in vitro*. Loss of RIOK2 leads to rapid death in full-body knockout mice as well as mice with RIOK2 loss specific to the hematopoietic system. Taken together, our results indicate that regulation of protein synthesis and ribosome biogenesis by RIOK2 is essential for the function of the hematopoietic system.

## Introduction

The precise regulation of protein synthesis is essential for the homeostasis of hematopoietic stem cells [1]. Ribosome levels play a central role in translational regulation and lineage commitment of the hematopoietic system [2]. Mutations in genes coding for ribosome biogenesis factors, ribosomal proteins as well as other components of the translational machinery are often associated with dysfunctional hematopoiesis and hematological diseases such as Diamond-Blackfan anemia, Schwachman Diamond syndrome and T-cell lymphoblastic leukemia [3, 4].

RIO kinase 2 (RIOK2) is an atypical kinase that has previously been reported to be involved in the export, assembly and maturation of 40S ribosomal proteins and therefore the regulation of mRNA translation to protein [5, 6]. The binding of RIOK2 to the pre-40S subunit has been shown to block premature translation initiation by preventing the binding of translation initiation factors to the pre-40S subunit. Haploinsufficiency of *Riok2* leads to decreased erythroid precursor frequency causing anemia [7], and several studies have highlighted a role of RIOK2 in various cancer types [8, 9]. A recent study identified RIOK2 as a transcriptional regulator of

novonordiskfonden.dk/en/ The work in the Helin laboratory (to K.H) was supported by: The Neye Foundation https://www.neye.dk/blog/neye-fonden/ The Kirsten and Freddy Johansens Foundation: https://www.kf-j.dk A center grant from the NNF to the NNF Center for Stem Cell Biology (NNF17CC0027852). novonordiskfonden.dk/en/ The Institute of Cancer Research. icr.ac.uk. None of the funders played a role in the study design, data collection and analysis, decision to publish or preparation of the manuscript.

**Competing interests:** K.H. is a consultant for Dania Therapeutics Aps and a scientific advisor for Hannibal Health Innovation. This does not alter our adherence to PLOS ONE policies on sharing data and materials. The other authors declare no competing interests.

key hematopoietic transcription factors such as GATA1, GATA2, SPI1, RUNX3 and KLF1 [10]. Recently, we have shown that RIOK2 is required for maintaining protein synthesis and ribosomal stability in leukemic cells, and we have therefore proposed RIOK2 as a potential therapeutic target for AML therapy [11].

To assess the suitability of RIOK2 as a potential target for hematological malignancies, it is critical to understand the functional consequences of RIOK2 loss on normal hematopoiesis. Here, we report on the consequences of RIOK2 loss on hematopoietic stem and progenitor cells as well as mature blood cells using two different conditional *Riok2* knockout mouse strains.

## Materials and methods

### Animal studies and mouse strains

*Riok2*$^{fl/fl}$; *Rosa26*::*Cre*$^{ERT2}$ and *Riok2*$^{fl/fl}$; *Mx1*::*Cre* mice were generated as previously described [11]. All mouse experiments conducted in Denmark were approved by the Danish Animal Ethical Committee (license number: 2017-15-0201-01176). All mouse experiments conducted in the UK were approved by the Animals in Science Regulation Unit (license number: PP5781054). All staff and animal technicians have the accreditation needed to conduct animal experiments in each country and are trained to keep animal welfare to a high standard.

### Competitive bone marrow transplantation and FACS analysis of the peripheral blood

B6-SJL mice were lethally irradiated (900 rad) followed by a transplantation of 50,000 *Riok2*$^{fl/fl}$; *Rosa26*::*Cre*$^{ERT2}$ or *Riok2*$^{fl/+}$; *Rosa26*::*Cre*$^{ERT2}$ cells mixed with wild-type cells in a 1:1 ratio via tail vein injection. Four mice were used in each group. The mice were shielded from one another for them not to see injections performed on all animals. Once injections with tamoxifen were started to induce Riok2 floxing animals were checked twice a day. Tamoxifen dissolved in corn oil was administered at 75 mg/kg bodyweight via intraperitoneal injection (using an ACUC approved injection procedure) once every 24 hours for a total of 5 consecutive days. FACS analysis of the peripheral blood was performed to determine the relative engraftment. Red blood cells were lysed using ammonium chloride solution (0.8% NH$_4$Cl) followed by a washing step using 3% FBS in PBS. Antibodies used for cell surface marker detection by FACS can be found in Table 1. FACS analysis was performed 4, 8, 12 and 16 weeks after transplantation and Tamoxifen injection.

### Non-competitive bone marrow transplantation and FACS analysis of the peripheral blood

B6-SJL mice were lethally irradiated (950 rad) and transplanted with 440,000 *Riok2*$^{fl/fl}$; *Rosa26*::*Cre*$^{ERT2}$ bone marrow cells via tail vein injection. Engraftment was confirmed by blood sampling 4 weeks post-transplant. Blood samples were lysed using RBC lysis buffer (Biolegend. Cat no: 420302) for 10 minutes, followed by a wash in 2% FBS in PBS. Antibodies used to determine the engraftment efficiency can be found in Table 1. Tamoxifen dissolved in corn oil was administered at 75 mg/kg bodyweight via intraperitoneal injection. Injection of 100 μL corn oil was used for control mice. Four animals were used in each group. The mice were shielded from one another for them not to see injections performed on all animals. Injections with tamoxifen or corn oil were given every day for a total of 5 consecutive days. Bone marrow was collected 10 days after the last tamoxifen or corn oil injections. Mouse wellbeing and survival was monitored daily after injection of polyIC or tamoxifen, respectively. Signs of illness,

**Table 1. FACS antibodies used in this study.**

| Antigen | Fluorophore | Manufacturer | Clone |
|---|---|---|---|
| c-KIT | PE-Cy7 | Invitrogen | 2B8 |
| CD45.1 | APC-Cy7 | BD Pharmigen | A20 |
| CD45.2 | AF700 | BD Pharmigen | 104 |
| Sca-1 | APC | Invitrogen | D7 |
| CD3e | PE-Cy5 | eBioscience | 145-2C11 |
| CD150 | BV650 | BioLegend | TC15-12F12.2 |
| CD48 | FITC | Invitrogen | HM4B-1 |
| CD16 | PE | eBioscience | 93 |
| CD32 | PE | eBioscience | 93 |
| Ter119 | PE | eBioscience | TER-119 |
| CD71 | APC | Invitrogen | R17217 |
| CD11b (Mac-1) | BV786 | BD Biosciences | M1/70 |
| Gr-1 | BV605 | BioLegend | RB6-8C5 |
| CD19 | BV650 | BD Horizon | 1D3 |
| CD4 | PE-Cy5 | Invitrogen | GK1.5 |
| CD8 | PE-Cy5 | Invitrogen | 53–6.7 |
| CD45.1 | PE | BioLegend | A20 |
| CD45.2 | BV421 | BioLegend | 104 |
| Sca-1 | SB436 | Invitrogen | D7 |
| CD48 | BV510 | BioLegend | HM48-1 |
| CD45.2 | FITC | BioLegend | 104 |
| CD135 | PE-CF594 | BD Biosciences | A2F10.1 |
| CD150 | PE-Cy7 | BioLegend | TC15-12F12.2 |
| CD34 | eFluor 660 | Invitrogen | RAM34 |
| CD16/32 | APC-R700 | BD Biosciences | 2.4G2 |
| c-KIT | APC-eFluor 780 | Invitrogen | 2B8 |

The lineage cocktail consisted of CD3e, Gr-1, CD11b, B220 and Ter119 (all PE-Cy5, BioLegend). In white are the antibodies used for the competitive transplant, and in blue the antibodies used for the non-competitive transplant.

including weight loss, bad appearance of fur, hunching, grimace scale and overall motility were assessed daily for each animal. All animals were euthanized immediately (within 5 minutes or less) upon visible signs of illness using cervical dislocation. No anesthesia or analgesia or methods to alleviate suffering were necessary during the experiments. No animals died before meeting the criteria for euthanasia.

## Whole body knockout of Riok2

6 to 12 weeks old $Riok2^{fl/fl}$; $Riok2^{fl/+}$; or $Riok^{+/+}$; $Rosa26$::$Cre^{ERT2}$ mice were injected with tamoxifen as described above in the non-competitive transplant setting. $Mx1$::$Cre$ mice were intraperitoneally injected with 0.4 mg Polyinosinic:polycytidylic acid (polyIC) dissolved in PBS. Four animals were used in each group. The injections were repeated every other day for a total of five injections over the course of 10 days. Mouse wellbeing and survival was monitored daily after injection of polyIC or tamoxifen, respectively. Signs of illness, including weight loss, bad appearance of fur, hunching, grimace scale and overall motility were assessed daily for each animal. All animals were euthanized immediately (within 5 minutes or less) upon visible signs of illness using cervical dislocation. No anesthesia or analgesia or methods to alleviate

suffering were necessary during the experiments. No animals died before meeting the criteria for euthanasia.

## Cell culture

LSK cells were sorted from *Riok2*<sup>fl/fl</sup>; *Rosa26*::*Cre*<sup>ERT2</sup> mice using a BD FACSAria III cell sorter (BD Biosciences) and subsequently cultured in X-VIVO 15 medium (Lonza, Cat.-Nr.: BE02-060F) containing 1% Pen/Strep, 2 mM L-Glutamine, 1% BSA, 0.1 mM beta-mercaptoethanol, 50 ng/ml mSCF, 10 ng/μl mIL-3, 50 ng/μl mIL-6.

## EdU labeling-based cell cycle analysis

LSK cells cultured in X-VIVO medium isolated from *Riok2*<sup>fl/fl</sup>; *ROSA26*::*Cre*<sup>ERT2</sup> and *Riok2*<sup>fl/+</sup>; *ROSA26*::*Cre*<sup>ERT2</sup> mice were treated with 10 μM 4-hydroxytamoxifen (OHT) or ethanol (EtOH) for 5 days after 48 hours in culture. EdU incorporation was measured using the Click-iT EdU Alexa Fluor 488 Flow Cytometry Assay Kit (Thermo Fisher, Cat.-Nr.: C10425) according to the manufacturer's instructions using 1 μM EdU and a labeling time of 45 minutes. Cells were stained with DAPI (1 μg/ml) before analysis. Flow cytometry was performed on a BD FACSAria III (BD Biosciences) and data analysis was performed using FlowJo software.

## Colony formation assay

Single-cell suspension of mouse bone marrow was enriched for c-KIT using CD117 microbeads (Miltenyi Biotech). Cells were stained with the indicated antibodies on ice for 30 minutes and subsequently sorted for the desired population on a BD FACSAria III sorter (BD Biosciences). Cells were plated into methylcellulose-based medium (M3534, StemCell Technologies) according to manufacturer's protocol at a density of 1,000 to 10,000 cells per ml. Colonies were counted manually on an inverted microscope with STEMgrid-6 and serial replating was performed in triplicates every week.

## Bone marrow FACS analysis

For the competitive transplantation experiments, femur, tibia and fibula were collected from both hind legs of the sacrificed mice. The bones were cleaned from flesh and crushed in a mortar under sterile conditions. The mortar was washed twice with 3 ml 3% FBS in PBS to collect all cells that were then filtered through a sterile 70 μm filter. After spinning, the cells were resuspended in 100 μl 3% FBS in PBS and 10 μl CD117 (c-KIT) beads (Mitenyi, Cat.-Nr: 130-091-224) and incubated for 15 minutes at 4°C. After washing, the cells were separated using the MACS separator system (Miltenyi Biotech) according to the manufacturer's instructions. Subsequently, the cells were counted and resuspended in the appropriate mix of antibodies for sorting or analysis.

For the non-competitive transplantation experiments, both hindlegs, hips and spine were harvested, cleaned, and crushed using a mortar and pestle. The mortar was washed in a total of 50 ml 3% FBS in PBS to collect all cells that were then filtered through a sterile 70 μm filter. After spinning, the samples were lysed using RBC lysis buffer (Biolegend. Cat no: 420302) for 3 min, followed by two washes in 3% FBS in PBS. Next, the cells were resuspended in 500 μl 3% FBS in PBS and 10 μl CD117 (c-KIT) beads (Mitenyi, Cat.-Nr: 130-091-224) and incubated for 20 minutes at 4°C. After washing, the cells were separated using the MACS separator system (Miltenyi Biotech) according to the manufacturer's instructions. Cells were counted and resuspended in a total volume of 100 μl for staining, followed by incubation with FACS antibodies for 90 minutes on ice. Samples were acquired on the CytoFLEX LX (Beckman Coulter),

and data were analysed using FlowJo. Absolute cell numbers were calculated based on the frequency of donor cells (CD45.2 positive) in each indicated population among single cells. The obtained frequency of single cells was then multiplied by the total number of isolated cells after processing the bone marrow to obtain the cell numbers displayed in Fig 1C.

The antibodies used for FACS analysis can be found in Table 1.

## Bone marrow histology

Tibia were dissected from the *Riok2*<sup>+/+</sup>, *Riok2*<sup>fl/+</sup>; *Rosa26*::Cre<sup>ERT2</sup> mice 20 days after tamoxifen treatment start or upon death for *Riok2*<sup>fl/fl</sup>; *Rosa26*::Cre<sup>ERT2</sup> mice. The bones were decalcified by incubation in 10% EDTA and fixed using 4% PFA. Processing, sectioning and HE staining of bones was performed using standard protocols [12].

## Genotyping

Genotyping was performed on DNA extracted from mouse tail tissue using the primer sequences shown in Table 2 using standard PCR protocols. Tail DNA was extracted by heating mouse tail tissue samples in 180 µl of 50 mM NaOH for 10 minutes at 95°C followed by the addition of 20 µl of 1M Tric-HCl (pH 8.0).

## Immunoblotting

500.000 LSK cells per treatment group were harvested and lysed in by the addition of 100 µl 1X LSB buffer to 300µl of PBS. The lysates were boiled at 95°C for 10 minutes. 50 µl of each sample were loaded on NuPAGE™ 4 to 12%, Bis-Tris, 1.0–1.5 mm, Mini Protein Gels (Thermo Fisher, Cat no: NP0322BOX). SDS-PAGE and blotting were performed according to standard protocols. Antibodies used for the immunoblotting are shown in Table 3.

## Results

To determine the role of RIOK2 in hematopoiesis, we performed a series of competitive and non-competitive bone marrow transplantation experiments. In the first approach, we mixed bone marrow cells from a previously established transgenic mouse line carrying floxed *Riok2* alleles [11] (S1A Fig) and a tamoxifen inducible Cre (Cre<sup>ERT2</sup>) expressed from the *Rosa26* locus (*Riok2*<sup>fl/fl</sup>; *Rosa26*::Cre<sup>ERT2</sup> and *Riok2*<sup>fl/+</sup>; *Rosa26*::Cre<sup>ERT2</sup>), and cells from a wild-type B6-SJL mouse in a 1:1 ratio. The mixed bone marrow cells were transplanted into lethally irradiated B6-SJL mice (Fig 1A). After verifying the contribution of *Riok2*<sup>fl/fl</sup>; *Rosa26*::Cre<sup>ERT2</sup> and *Riok2*<sup>fl/+</sup>; *Rosa26*::Cre<sup>ERT2</sup> cells to different lineages of the peripheral blood at 4 weeks post-transplantation (S1B Fig), we injected tamoxifen intraperitoneally to induce the recombination of the *Riok2* locus [13] (S1C Fig) We measured the contribution of *Riok2*<sup>fl/fl</sup>; *Rosa26*::Cre<sup>ERT2</sup> and *Riok2*<sup>fl/+</sup>; *Rosa26*::Cre<sup>ERT2</sup> bone marrow cells (CD45.2<sup>+</sup>) to the peripheral blood 4, 8,12 and 16 weeks after transplantation (Fig 1B). We observed a loss of both myeloid and lymphoid CD45.2<sup>+</sup> cells in the transplanted mice (Fig 1B). In contrast, heterozygous loss of *Riok2* did not lead to significant changes in the analyzed hematopoietic cells (Fig 1B). These results suggest that RIOK2 is essential for the maintenance of mature peripheral blood cells.

To investigate the consequences of RIOK2 loss on hematopoietic stem and progenitor cells, we performed a non-competitive bone marrow transplant where we used bone marrow collected from *Riok2*<sup>fl/fl</sup>; *Rosa26*::Cre<sup>ERT2</sup> mice as donor cells and B6-SJL mice as recipients. After validation of successful engraftment (S1D Fig), we performed tamoxifen injections in half of the transplanted mice. The other half received corn oil injections as a control. To assess the effect of RIOK2 loss on different stem and progenitor cell types, we collected bone marrow

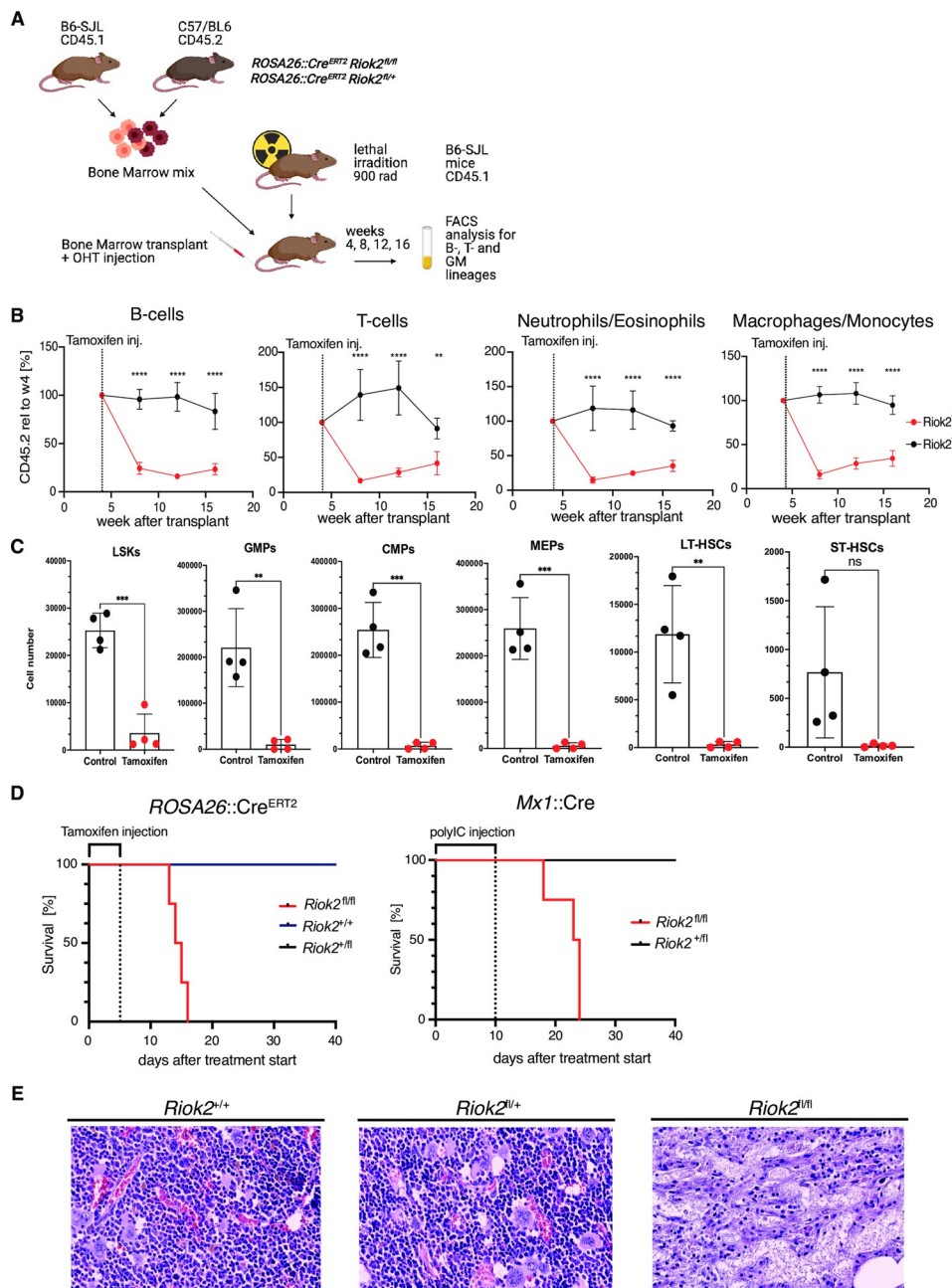

**Fig 1. RIOK2 is essential for the proliferation of mature blood cells *in vivo* and for mouse survival. A)** Schematic overview of the competitive bone marrow transplantation. Lethally irradiated B6/SJL mice were injected with a mix of bone marrow cells from *Riok2*^fl/fl^; *Rosa26::Cre*^ERT2^ or *Riok2*^fl/+^; *Rosa26::Cre*^ERT2^ mice and cells from wild-type B6-SJL mice in a 1:1 ratio. 4 weeks after transplantation, all animals were injected intraperitoneally with tamoxifen to induce Cre-mediated recombination of the *Riok2* locus. Created with Biorender.com. **B)** CD45.2 chimerism in the indicated peripheral blood cell types 4, 8, 12 and 16 weeks after bone marrow transplantation. Data is represented as mean ± standard deviation (SD). n = 4 animals per group. **: p<0.01; ***: p<0.001 by Unpaired t-test. **C)** Bar graphs depicting the absolute number of CD45.2 positive cells in each indicated population of mice transplanted with *Riok2*^fl/fl^; *Rosa26::Cre*^ERT2^ treated with corn oil (control) or tamoxifen. Data is represented as mean ± standard deviation (SD). n = 4 animals per group. **: p<0.01; ****: p<0.0001 by 2way ANOVA test followed by Tukey's multiple comparison test. **D)** Kaplan-Meier survival curves of the indicated transgenic mouse strains after deletion of *Riok2* using tamoxifen (left panel) or polyIC (right panel). n = 4 animals per group. **E)** Bone marrow histology revealed by HE-staining of bone marrow from *Riok2*^+/+^, *Riok2*^fl/+^, or *Riok2*^fl/fl^; *Rosa26::Cre*^ERT2^ mice collected 20 days after tamoxifen treatment or upon death. Scale bar represents 100 μm.

**Table 2. Sequences for PCR primers used in this study.**

| Primer name | Sequence (5'-3') |
| --- | --- |
| Riok_mouse_Common_en2_F_out | ACTTCTTACGCCAGGAACCT |
| Riok_mouse_Common_en2_R | CCAACTGACCTTGGGCAAGAACAT |
| Common_LoxP_F | GAGATGGCGCAACGCAATTAAT |
| Common_LoxP_R_outside | TGCTTGAATAAATGGCTCCCTG |
| Common_3'_F | CACACCTCCCCCTGAACCTGAAA |
| Flp_KOMP_rev | CTTTTGGAAGAGCAGTCAGG |

cells 10 days after the last injections and performed flow cytometry, as the mice started to develop symptoms of bone marrow failure. The result of this analysis showed that RIOK2 loss led to a significant decrease of all stem and progenitor cell types, including LSK, GMP, CMP, MEP, LT-HSC and ST-HSC populations (Figs 1C and S1E and S1F). This result indicates that in addition to being essential for peripheral blood cells, RIOK2 is also essential for the maintenance of hematopoietic stem and progenitor cells in the bone marrow.

To assess the consequence of RIOK2 loss in adult mice, we treated *Riok2*$^{fl/fl}$; *Rosa26*:: *Cre*$^{ERT2}$, *Riok2*$^{fl/+}$; *Rosa26*::*Cre*$^{ERT2}$, and *Riok2*$^{+/+}$; *Rosa26*::*Cre*$^{ERT2}$ mice with tamoxifen. Whereas the loss of both alleles of *Riok2* led to rapid death of the mice with a median survival of 14.5 days, mice expressing one allele or control mice were not affected by tamoxifen treatment (Fig 1D, left panel). Inducing RIOK2 loss specifically in the hematopoietic system using a *Mx1*::*Cre* driven recombination of the *Riok2* allele also led to the rapid death of the mice with a median survival of 23.5 days (Fig 1D, right panel). To investigate the consequences of RIOK2 loss on the bone marrow morphology, we collected tibia from dead mice and performed bone marrow histology. We observed a significant loss of bone marrow cellularity in *Riok2*$^{fl/fl}$; *Rosa26*::*Cre*$^{ERT2}$ but not in *Riok2*$^{+/+}$ or *Riok2*$^{fl/+}$ mice, indicating that RIOK2 is essential for cellular homeostasis in the bone marrow (Fig 1E).

To further investigate the phenotypic consequences of RIOK2 loss, we investigated the effects of its depletion on differentiation, self-renewal and proliferation of HSPCs *in vitro* (Fig 2A). For this, we sorted LSK cells isolated from *Riok2*$^{fl/fl}$; *Rosa26*::*Cre*$^{ERT2}$ and *Riok2*$^{fl/+}$; *Rosa26*::Cre$^{ERT2}$ mice, which had been treated with 4-hydroxytamoxifen (OHT) prior to sorting. As shown in S2A Fig, this led to the decrease of RIOK2 protein levels in the OHT-treated LSK cells. The abrogation of RIOK2 expression led to a significant increase in the relative number of Lin$^-$ and LSK cells (Figs 2B and S2B), potentially indicating that more mature hematopoietic cell types are more sensitive to the acute loss of RIOK2. Moreover, *Riok2* knockout cells showed a strong decrease in the capacity to form colonies in methylcellulose (Fig 2C). These results further support our data showing that RIOK2 is required for hematopoietic stem cell maintenance.

Lastly, to determine the effects of RIOK2 loss on cell proliferation in more detail, we performed a FACS-based EdU labeling assay on LSK cells. This analysis showed that deletion of *Riok2* led to cell cycle arrest and apoptosis in *Riok2*$^{fl/fl}$ LSK cells as indicated by a strong increase in the sub-G1 fraction (Fig 2D).

**Table 3. Western Blot antibodies used in this study.**

| Epitope | Manufacturer | Application | Dilution | Cat.-No. |
| --- | --- | --- | --- | --- |
| Mouse RIOK2 | Genscript (custom order) | Western Blot | 1:4000 | - |
| β-tubulin | abcam | Western Blot | 1:5000 | ab15568 |

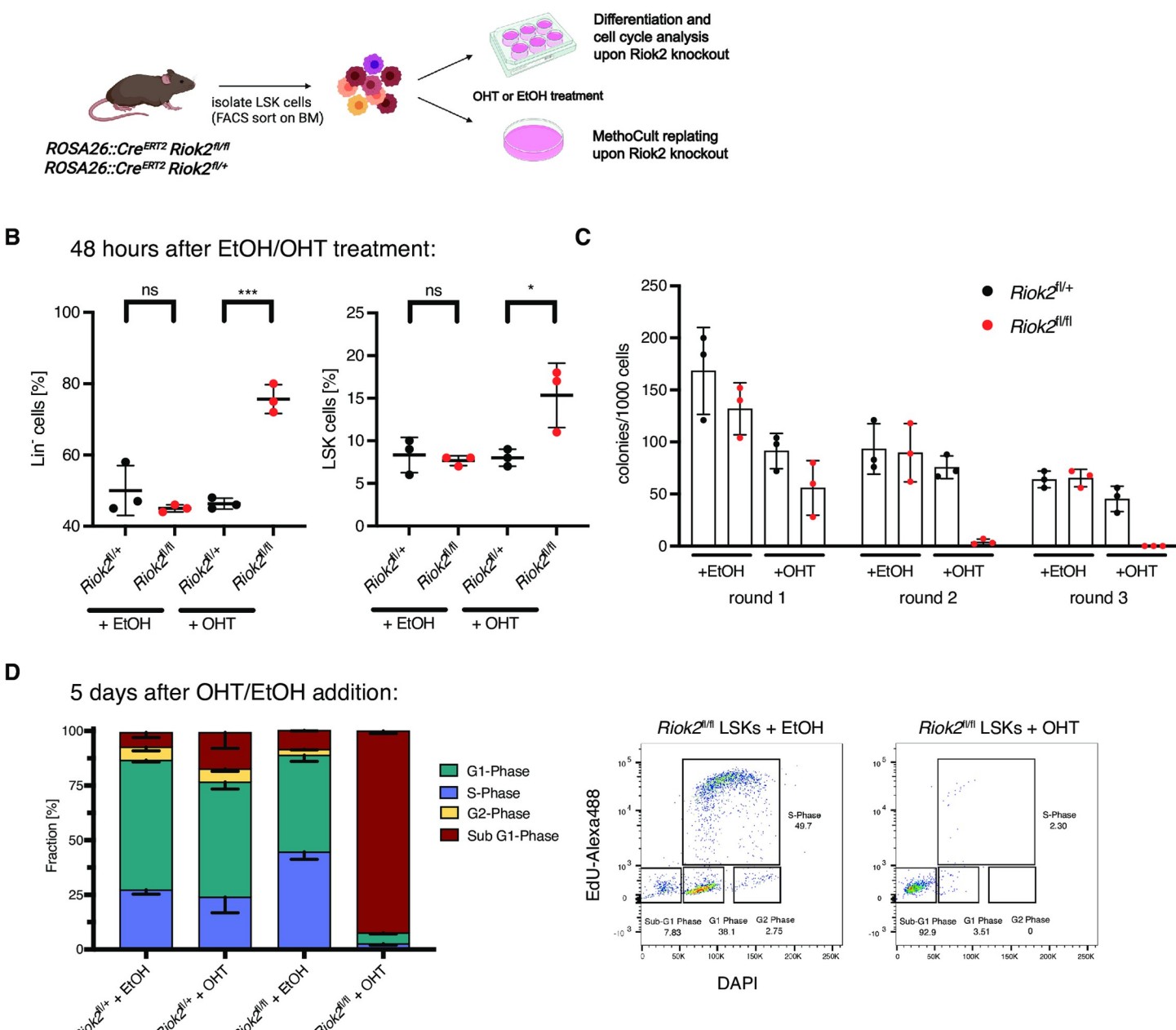

**Fig 2. RIOK2 loss affects differentiation and proliferation of hematopoietic stem and progenitor cells *in vitro*. A)** Experimental outline to determine the effects of RIOK2 loss on hematopoietic stem and progenitor cells *in vitro*. Created with Biorender.com. **B)** Percentage of Lin⁻ cells and LSK cells 48h after the addition of either ethanol (EtOH) or 4-hydroxytamoxifen (OHT) to the culture medium of *Riok2*fl/fl; *Rosa26*::Cre*ERT2* and *Riok2*fl/+; *Rosa26*::Cre*ERT2* cells. Data is represented as mean ± standard deviation (SD). n = 3 biological replicates per group. A Student's t-test was performed to assess statistical significance (ns = not significant, \*$p < 0.05$, \*\*\*$p < 0.001$). **C)** MethoCult replating assay using LSK cells sorted from bone marrow of the indicated genotypes. Cells were replated once per week. Data is represented as mean ± standard deviation (SD). n = 3 biological replicates per group. **D)** Left panel: EdU labeling of *Riok2*fl/fl; *Rosa26*::Cre*ERT2* and *Riok2*fl/+; *Rosa26*::Cre*ERT2* LSK cells. Error bars represent standard deviation (SD), n = 3 biological replicates. Right panel: representative FACS plot of *Riok2*fl/fl treated with either EtOH or OHT for 5 days.

## Discussion

We have addressed the phenotypic consequences of RIOK2 loss on the hematopoietic system and mouse survival. Collectively, our data suggests that RIOK2 is required for the expansion,

homeostasis and differentiation of hematopoietic stem and progenitor cells as well as for the homeostasis of lymphoid and myeloid cells in the peripheral blood. These results are in agreement with recent data showing a key role for RIOK2 in the transcriptional regulation of blood cell development [10]. Additionally, loss of RIOK2 strongly affects hematopoietic stem and progenitor cell survival *in vivo* as well as the differentiation, self-renewal and proliferation of HSPCs *in vitro*. We have also observed a relative increase in Lin- as well as LSK cells upon loss of RIOK2 *in vitro*, which may be because more mature hematopoietic cells have higher rates of protein synthesis [1] and thereby are more sensitive to RIOK2 loss. Recently, we showed that loss of RIOK2 in mouse embryonic fibroblasts leads to a stop in proliferation and induction of senescence, indicating that RIOK2 is not required for cell survival per se [11]. While the results lower the attractiveness of pursuing RIOK2 inhibition as a pharmacological target, several inhibitors of ribosome biogenesis and protein synthesis are currently in clinical trials, such as the ribosome biogenesis inhibitor CX-5461 [14]. Further investigations will be necessary to understand the tissue specific dependency of RIOK2.

In summary, we have demonstrated that RIOK2 is required for homeostatic maintenance of the hematopoietic system as well as hematopoietic stem and progenitor cells. Further studies are required to investigate if there is a therapeutic window allowing for the safe and efficient targeting of RIOK2 in cancer.

## Supporting information

**S1 Fig.** A) Schematic overview of the Riok2fl/fl locus. Exons 4 and 5 are flanked by LoxP site that are excised after the induction of Cre recombinase. B) Representative FACS plots showing the engraftment of Riok2fl/fl and Riok2fl/+ B-cells 4 weeks after transplantation using CD45.1/CD45.2 staining. C) Agarose gel showing the genotyping of the transplanted bone marrow of ROSA26::CreERT2, Riok2fl/+ and Riok2fl/fl mice used in the competitive bone marrow transplantation assay after termination of the experiment. Arrows indicate wild-type and floxed as well as recombined and non-recombined alleles. The lanes between the bottom and top gels are matched and represent the same mice. D) Bar graph depicting the percentage of CD45.1 or CD45.2 positive cells in lethally irradiated mice transplanted with Riok2fl/fl; Rosa26::CreERT2. Data is represented as mean ± standard deviation (SD) (n = 8). ****: p<0.0001 by Unpaired t-test. E) Gating strategy for HSPC population identification and representative result for mice injected with corn oil. F) Gating strategy for HSPC population identification and representative result for mice injected with tamoxifen and experiencing symptoms of bone marrow failure.
(TIF)

**S2 Fig.** A) Immunoblot showing RIOK2 protein levels in LSK cells cultured in X-VIVO medium from the indicated genotypes 48 hours after the addition of either EtOH or OHT to the culture medium. B) Representative FACS plots for X-VIVO cultured LSK cells 48 hours after the addition of EtOH or OHT to the culture medium.
(TIF)

**S1 Raw images.**
(PDF)

**S1 File.**
(XLSX)

## Acknowledgments

We thank members of the Helin Lab for discussions and Mafalda Araujo Pereira for suggestions for the FACS analysis. Figs 1A and 2A were created with Biorender.com.

## Author Contributions

**Conceptualization:** Jan-Erik Messling, Kristian Helin.

**Data curation:** Isabel Peña-Rømer.

**Formal analysis:** Jan-Erik Messling, Isabel Peña-Rømer, Ann Sophie Moroni, Sarah Bruestl.

**Funding acquisition:** Kristian Helin.

**Investigation:** Jan-Erik Messling, Isabel Peña-Rømer, Ann Sophie Moroni, Sarah Bruestl, Kristian Helin.

**Methodology:** Jan-Erik Messling, Isabel Peña-Rømer, Ann Sophie Moroni, Sarah Bruestl.

**Supervision:** Kristian Helin.

**Validation:** Jan-Erik Messling, Isabel Peña-Rømer, Ann Sophie Moroni, Sarah Bruestl.

**Visualization:** Jan-Erik Messling, Isabel Peña-Rømer, Ann Sophie Moroni.

**Writing – original draft:** Jan-Erik Messling, Kristian Helin.

**Writing – review & editing:** Isabel Peña-Rømer, Ann Sophie Moroni, Sarah Bruestl.

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
