## [Decision Letter · Decision Letter 0]

8 Jan 2024

PONE-D-23-40786RIO-kinase 2 is essential for hematopoiesisPLOS ONE

Dear Dr. Helin,

Thank you for submitting your manuscript to PLOS ONE. After careful consideration, we feel that it has merit but does not fully meet PLOS ONE’s publication criteria as it currently stands. Therefore, we invite you to submit a revised version of the manuscript that addresses the points raised during the review process.

We look forward to receiving your revised manuscript.

Kind regards,

Jimin Han

Academic Editor

PLOS ONE

[K.H. is a consultant for Dania Therapeutics Aps and a scientific advisor for Hannibal Health Innovation. The other authors declare no competing interests.]. 

5. We note that Figure 1A, 1E, 2A and 2E in your submission contain copyrighted images. All PLOS content is published under the Creative Commons Attribution License (CC BY 4.0), which means that the manuscript, images, and Supporting Information files will be freely available online, and any third party is permitted to access, download, copy, distribute, and use these materials in any way, even commercially, with proper attribution. For more information, see our copyright guidelines: http://journals.plos.org/plosone/s/licenses-and-copyright.

a. You may seek permission from the original copyright holder of Figure 1A, 1E, 2A and 2E to publish the content specifically under the CC BY 4.0 license. 

Additional Editor Comments:

Dear Dr. Kristian Helin

This research paper, titled " RIO-kinase 2 is essential for hematopoiesis" explores the role of RIO-kinase 2 (RIOK2) in hematopoiesis, the process of blood cell formation. The study uses conditional knockout mouse models to demonstrate that RIOK2 is crucial for the differentiation and maintenance of hematopoietic stem and progenitor cells, as well as for the survival of fully differentiated blood cells. The absence of RIOK2 leads to rapid death in mice, indicating its vital role in the hematopoietic system. This research provides insights into the potential targeting of RIOK2 for treating blood cell-related disorders. The whole story is quite in-depth and it’s very interesting. I'm including feedback from two expert reviewers regarding your manuscript. Their insights highlight the study's potential while also presenting various critiques and recommendations. After read reviewer’s feedback, I believe this article requires careful revision. So, I decide give you major revision decision.

It's our hope that you find this feedback constructive. Should additional data be available to address the raised concerns, we are open to examining a modified submission. Our timeline is adaptable for any necessary extended research. For submissions linked to a specific thematic issue, please liaise with the editorial team concerning resubmission deadlines. We emphasize that key findings must be backed by comprehensive statistics from ample independent trials. Manuscripts may be returned at the editor's discretion for the completion of any lacking or partial statistical data.

I strongly suggest rearranging the figures, so they are more reader-friendly and more clearly. For more details, please refer to the layout instructions of PLOS ONE.

Yours sincerely,

Jimin Han

Academic Editor,

PLOS ONE

Review Comments to the Author

Reviewer 1:

The study explores the role of RIOK2 in blood cell formation, both in vivo and in vitro, with the manuscript conveying an evident phenotype associated with the knockout of RIOK2.

Major Point:

The figures included in the main text are of such low quality that it is difficult to discern the text within the images. Enhancement of image clarity is necessary.

Minor Points:

1. In Fig S1C, concerning the lower image of genotyping, it is unclear what the genotype is for the control. Why do two lanes of ctrl display mismatched bands?

2. Is it critical to establish the baseline levels of CD45.1 and CD45.2 in mice before the transplantation process? Additionally, would it enhance clarity to more distinctly show the reduction of CD45+ cells following radiation?

3. Some of the statistical charts, like those in Figure 1B, 1C, and FigS1D, would benefit from an analysis of significance to support the findings.

Reviewer 2:

RIOK2 is important for hematopoietic stem cell（HPSC）maintance and differentiation. This manuscript investigates the effects of RIOK2 on HPSC function and HPSC transplantation using a conditional knockout mouse model in the hematopoietic system. The research findings have significant implications for understanding HPSC function and transplantation. However, revisions need to be made to the manuscript before it can be accepted.

In Figure 1, we propose that both Riok2fl/fl; Rosa26::CreERT2 and Riok2fl/+; Rosa26::CreERT2 have the effect of downregulating RIOK2 gene expression. Why was Riok2+/+; Rosa26::CreERT2 mice not used as a control?

In Supplementary Figure 1, should there be additional images demonstrating the efficiency of RIOK2 gene knockdown in Riok2fl/fl; Rosa26::CreERT2 and Riok2fl/+; Rosa26::CreERT2 mice, rather than solely relying on DNA-level identification?

In Figure 2, the experimental data regarding the impact of Riok2 knockout on HSC proliferation and differentiation is insufficient to support the authors' conclusions. Are there additional evidence available to substantiate these claims?

Reviewers' comments:

Reviewer's Responses to Questions

**Comments to the Author**

1. Is the manuscript technically sound, and do the data support the conclusions?

Reviewer #1: Yes

Reviewer #2: Yes

2. Has the statistical analysis been performed appropriately and rigorously? 

Reviewer #1: Yes

Reviewer #2: Yes

3. Have the authors made all data underlying the findings in their manuscript fully available?

Reviewer #1: Yes

Reviewer #2: Yes

4. Is the manuscript presented in an intelligible fashion and written in standard English?

Reviewer #1: Yes

Reviewer #2: Yes

5. Review Comments to the Author

Reviewer #1: 

RIOK2 is important for hematopoietic stem cell（HPSC）maintance and differentiation. This manuscript investigates the effects of RIOK2 on HPSC function and HPSC transplantation using a conditional knockout mouse model in the hematopoietic system. The research findings have significant implications for understanding HPSC function and transplantation. However, revisions need to be made to the manuscript before it can be accepted.

In Figure 1, we propose that both Riok2fl/fl; Rosa26::CreERT2 and Riok2fl/+; Rosa26::CreERT2 have the effect of downregulating RIOK2 gene expression. Why was Riok2+/+; Rosa26::CreERT2 mice not used as a control?

In Supplementary Figure 1, should there be additional images demonstrating the efficiency of RIOK2 gene knockdown in Riok2fl/fl; Rosa26::CreERT2 and Riok2fl/+; Rosa26::CreERT2 mice, rather than solely relying on DNA-level identification?

In Figure 2, the experimental data regarding the impact of Riok2 knockout on HSC proliferation and differentiation is insufficient to support the authors' conclusions. Are there additional evidence available to substantiate these claims?

Reviewer #2: The study explores the role of RIOK2 in blood cell formation, both in vivo and in vitro, with the manuscript conveying an evident phenotype associated with the knockout of RIOK2.

Major Point:

The figures included in the main text are of such low quality that it is difficult to discern the text within the images. Enhancement of image clarity is necessary.

Minor Points:

1. In Fig S1C, concerning the lower image of genotyping, it is unclear what the genotype is for the control. Why do two lanes of ctrl display mismatched bands?

2. Is it critical to establish the baseline levels of CD45.1 and CD45.2 in mice before the transplantation process? Additionally, would it enhance clarity to more distinctly show the reduction of CD45+ cells following radiation?

3. Some of the statistical charts, like those in Figure 1B, 1C, and FigS1D, would benefit from an analysis of significance to support the findings.

6. PLOS authors have the option to publish the peer review history of their article (what does this mean?). If published, this will include your full peer review and any attached files.

Reviewer #1: **Yes: **Jia He

Reviewer #2: No

---

## [Author Response · Author response to Decision Letter 0]

19 Feb 2024

Journal requirements

We have adapted the manuscript to the style requirements.

As requested, we have added additional information to the Methods section. 

[K.H. is a consultant for Dania Therapeutics Aps and a scientific advisor for Hannibal Health Innovation. The other authors declare no competing interests.]. 

We have added additional information to the Competing Interests section. 

The competing Interests statement now reads:

K.H. is a consultant for Dania Therapeutics Aps and a scientific advisor for Hannibal Health Innovation. This does not alter our adherence to PLOS ONE policies on sharing data and materials. The other authors declare no competing interests.

We have included Supporting Information that include the original underlying images for DNA agarose gels and Western blot.

5. We note that Figure 1A, 1E, 2A and 2E in your submission contain copyrighted images. All PLOS content is published under the Creative Commons Attribution License (CC BY 4.0), which means that the manuscript, images, and Supporting Information files will be freely available online, and any third party is permitted to access, download, copy, distribute, and use these materials in any way, even commercially, with proper attribution. For more information, see our copyright guidelines: http://journals.plos.org/plosone/s/licenses-and-copyright.

a. You may seek permission from the original copyright holder of Figure 1A, 1E, 2A and 2E to publish the content specifically under the CC BY 4.0 license. 

 We have an academic license from Biorender that gives us the permission to use the figures in academic publications (https://www.biorender.com/academic-license). We have acknowledged Biorender as a source in the appropriate figures. 

Response to Reviewers

Reviewer #1:

In Figure 1, we propose that both Riok2fl/fl; Rosa26::CreERT2 and Riok2fl/+; Rosa26::CreERT2 have the effect of downregulating RIOK2 gene expression. Why was Riok2+/+; Rosa26::CreERT2 mice not used as a control?

Heterozygous deletion of Riok2 does not influence mature hematopoietic cells in our competitive transplants. Figure 1B shows that the CD45.1:CD45.2 ratio is stable across all timepoints and for all cell types in the Riok2fl/+ competitive experiments. Additionally, as shown in Figure 1 D and E, we do not observe differences in mouse survival and bone marrow composition upon loss of one allele of Riok2 compared to wild-type mice. Therefore, we used Riok2fl/+ as a control for the effect of deleting both alleles of Riok2. 

In Supplementary Figure 1, should there be additional images demonstrating the efficiency of RIOK2 gene knockdown in Riok2fl/fl; Rosa26::CreERT2 and Riok2fl/+; Rosa26::CreERT2 mice, rather than solely relying on DNA-level identification?

We agree with the reviewer that it would be nice to show. However, unfortunately, it is not possible to obtain enough cells to perform immunoblotting of RIOK2 on material from the competitive transplant, because the RIOK2 knockout cells were rapidly outcompeted by the non-recombined wild-type cells. Importantly, we have shown in supplemental Figure 2A that tamoxifen treatment leads to a rapid and efficient depletion of RIOK2 protein on hematopoietic stem- and progenitor cells, supporting the efficiency of the deletion of both alleles of Riok2.

In Figure 2, the experimental data regarding the impact of Riok2 knockout on HSC proliferation and differentiation is insufficient to support the authors' conclusions. Are there additional evidence available to substantiate these claims?

We agree with the author that the claims were insufficient and have removed the model that concluded independent mechanisms and a differentiation block for hematopoietic stem- and progenitor cells and mature cells. 

Reviewer #2: 

The figures included in the main text are of such low quality that it is difficult to discern the text within the images. Enhancement of image clarity is necessary.

We are sorry for this. The link provided in the downloaded pdf of the manuscript should have provided a high-resolution TIFF file of each of the figures. We have ensured that the figures in the revised version agree with the requirement of the journal. 

Minor Points:

1. In Fig S1C, concerning the lower image of genotyping, it is unclear what the genotype is for the control. Why do two lanes of ctrl display mismatched bands?

The control bands are a negative (wild-type) control, which is the non-recombined locus as well as a positive (Riok2fl/fl) control. We have added additional description to the supplemental Figure 1C and its figure legend to enhance clarity.

2. Is it critical to establish the baseline levels of CD45.1 and CD45.2 in mice before the transplantation process? Additionally, would it enhance clarity to more distinctly show the reduction of CD45+ cells following radiation?

We have used appropriate controls to correctly set up the FACS gates for determination of CD45.1 and CD45.2 levels. In competitive transplantation it is not common practise to establish baseline levels. The assay is to establish the competition and not the total levels of each population. 

3. Some of the statistical charts, like those in Figure 1B, 1C, and FigS1D, would benefit from an analysis of significance to support the findings.

We have performed additional statistical analyses for the figures requested by the reviewer and added appropriate descriptions to the figures and figure legends.

---

## [Decision Letter · Decision Letter 1]

4 Mar 2024

RIO-kinase 2 is essential for hematopoiesis

PONE-D-23-40786R1

Dear Dr. Helin,

This time the authors have corrected it in such detail that I think PLOS ONE is ready to accept the paper.I asked the Reviewer to review it, and none of them had any opinion.I firmly believe that this article will contribute to the advancement of this field.

We’re pleased to inform you that your manuscript has been judged scientifically suitable for publication and will be formally accepted for publication once it meets all outstanding technical requirements.

Kind regards,

Jimin Han

Academic Editor

PLOS ONE

Reviewers' comments:

Reviewer's Responses to Questions

**Comments to the Author**

1. If the authors have adequately addressed your comments raised in a previous round of review and you feel that this manuscript is now acceptable for publication, you may indicate that here to bypass the “Comments to the Author” section, enter your conflict of interest statement in the “Confidential to Editor” section, and submit your "Accept" recommendation.

Reviewer #3: All comments have been addressed

2. Is the manuscript technically sound, and do the data support the conclusions?

Reviewer #3: Yes

3. Has the statistical analysis been performed appropriately and rigorously? 

Reviewer #3: Yes

4. Have the authors made all data underlying the findings in their manuscript fully available?

Reviewer #3: Yes

5. Is the manuscript presented in an intelligible fashion and written in standard English?

Reviewer #3: Yes

6. Review Comments to the Author

Reviewer #3: (No Response)

7. PLOS authors have the option to publish the peer review history of their article (what does this mean?). If published, this will include your full peer review and any attached files.

Reviewer #3: No

---

## [Editor Report · Acceptance letter]

23 Mar 2024

PONE-D-23-40786R1 

PLOS ONE

Dear Dr. Helin, 

I'm pleased to inform you that your manuscript has been deemed suitable for publication in PLOS ONE. Congratulations! Your manuscript is now being handed over to our production team.

Kind regards, 

on behalf of

Dr. Jimin Han 

Academic Editor

PLOS ONE